# On the Choice of Different Water Model in Molecular Dynamics Simulations of Nanopore Transport Phenomena

**DOI:** 10.3390/membranes12111109

**Published:** 2022-11-07

**Authors:** Chulwoo Park, Ferlin Robinson, Daejoong Kim

**Affiliations:** Department of Mechanical Engineering, Sogang University, Seoul 04107, Korea

**Keywords:** carbon, graphene, multilayered graphene, water transport, water model, nanopore membrane

## Abstract

The water transport through nanoporous multilayered graphene at 300k is investigated using molecular dynamics (MD) simulation with different water models in this study. We used functionalized and non-functionalized membranes along with five different 3-point rigid water models: SPC (simple point charge), SPC/E (extended simple point charge), TIP3P-FB (transferable intermolecular potential with 3 points—Force Balance), TIP3P-EW (transferable intermolecular potential with 3 points with Ewald summation) and OPC3 (3-point optimal point charge) water models. Based on our simulations with two water reservoirs and a porous multilayered graphene membrane in-between them, it is evident that the water transport varies significantly depending on the water model used, which is in good agreement with previous works. This study contributes to the selection of a water model for molecular dynamics simulations of water transport through multilayered porous graphene.

## 1. Introduction

The dynamic view of the microscopic systems can be studied with the help of molecular dynamics (MD) simulations. The study of mass transport through nanoscale channels has gained much interest in recent times as it possesses a large application potential. Nanoscale materials such as nanotubes, nanopores and nanogaps have shown promising potential in the desalination process [1,2,3,4,5,6,7]. These nanoscale materials show better ion selectivity, high efficiency and low cost. Membrane desalination is considered to be more energy efficient than thermal desalination methods [8]. Additionally, the membrane desalination technique known as reverse osmosis (RO) is considered to be more environmentally friendly. The water flux across a membrane scales inversely to the thickness of the membrane. Hence, ultra-thin membranes promise better water transport. Graphene consists of sp2 carbon atoms arranged in a hexagonal honeycomb lattice, making it the ultimate ultrathin material. It also exhibits excellent mechanical, thermal and electronic properties [9]. Hence, the use of graphene in the desalination process has grabbed significant attention.

Water transport through single-layer graphene using molecular dynamics was first shown by Suk et al. [10]. Single-layer graphene membrane with hydrogenated and hydroxylated nanopores has shown fast water transport along with excellent salt rejection [8]. It has been reported that the desalination performances of pyridinic nitrogen-doped membranes exhibit higher water flux several orders of magnitude higher than polymeric reverse osmosis (RO) membranes [11]. The removal of heavy metal ions from water using functionalized graphene has been demonstrated using a molecular dynamics study [12]. In an experimental work conducted in 2015 to study the ionic transport through hydrophobic nanopores, the researchers found that there is a constant weak surface charge density for nanopore diameters greater than 3 nm [13].

Water is one of the most plentiful substances on planet earth and it is one of the most extensively studied substances [14,15]. Even though numerous studies have been carried out about it, our understanding of its behavior and its distinctive properties are incomplete [16]. To study water molecule and their interaction with other substances using molecular dynamics require accurate water models. Most of the studies in the MD use fixed-charge rigid non-polarizable water models due to their computational efficiency [17]. Rigid 3-point water models are the commonly used water models in atomistic MD simulations as they can reproduce many properties of water. Rigid 3-point water models represent water molecules as triatomic molecules with rigid bonds. simple point charge (SPC) [18], simple point charge-extended (SPC/E) [19] and transferable intermolecular potential 3P (TIP3P) [20] are the popular rigid 3-point water models. SPC water model is the oldest water model that is commonly in use today [21]. The 3-point optimal point charge (OPC3) model [17] is one of the new rigid 3-point water models which is comparable to or significantly better for simulations of divalent metal ions than the other 3-point water models [22].

Using molecular dynamics, numerous studies have been conducted to study the water transport properties of multilayered nanoporous graphene. In most of these simulation studies, water models such as transferable intermolecular potential 3 point (TIP3P) [23,24,25,26] and simple point charge-extended (SPC/E) [27,28,29,30,31] are widely used. In an MD study conducted in 2010, the interaction of the water molecules confined in between two graphene sheets showed that the results are qualitatively and semi-qualitatively equivalent for the TIP3P and SPC/E water models [32]. In a recent MD study in early 2022 carried out on water transport through carbon nanotubes, researches showed that the water models do not have any influence on the conductance trend as the diameter of the tube is increased [33]. In a study published in 2020, rotating graphene membranes were found to have almost 100% salt rejection, even with pores larger than hydrated ions, and to have high water permeability and ultra-selectivity simultaneously [34]. According to a recent study, fabricating distillation membranes with vertically aligned channels with a hydrophobicity gradient can be accomplished by removing imine bonds from covalent organic framework films, resulting in a threefold increase in water permeance over state-of-the-art membrane distillation [35]. The influence of water models on desalination performance in the single-layered system showed that the water flux variations are as high as 84% among different water models [36]. It appears that there has been no systematic study carried out on how the different water models influence water transport through multilayered nanoporous graphene.

In this study, we have performed molecular dynamics simulations on functionalized and non-functionalized nanoporous multilayered graphene using different water models. SPC (simple point charge) [18], SPC/E (extended simple point charge) [19], TIP3P-FB (transferable intermolecular potential with 3 points—Force Balance) [37], TIP3P-EW (transferable intermolecular potential with 3 points with Ewald summation) [38] and OPC3 (3-point optimal point charge) [17] are the different water models used in this study. The aim of this study is to find out the amount of variation in the number of water molecules transported across multilayered graphene, both functionalized and non-functionalized, using different water models. It also gives an overall picture of the key attributes that lead to the variation of the number of water molecules transported through the multilayered porous graphene membrane.

## 2. Methods and Model

The nanoporous multilayered graphene used in this study is modeled in SAMSON (Software for Adaptive Modeling and Simulation of Nanosystems) software [39]. Minimization of the energy of the structure is carried out using the FIRE (Fast Inertial Relaxation Engine) algorithm [40]. The nanoporous multilayered graphene system consists of 5 layers of nanoporous graphene. These graphene layers are heavily stacked with an interlayer distance between the graphene sheets of 3.5 Å. The dimensions of the graphene sheets are 30 Å × 30 Å. Water molecules are placed on both sides of the nanoporous multilayered graphene structure. The simulation setup used in this study is illustrated in Figure 1.

The simulation box (30 Å × 30 Å × 160 Å) has a total of 2138 water molecules, of which 1710 water molecules are in the feed region and 428 water molecules in the permeate region. SPC, SPC/E, TIP3P-FB, TIP3P-EW and OPC3 are the different water models used in this study. The configurations of these water models are given in Figure 2.

The force field parameters of the water models used in this study are listed in Table 1.

AIREBO (adaptive intermolecular reactive bond order) potential [41] is used for the carbon atoms and the functionalized hydrogen atoms of the porous membrane. We used the Lorentz–Berthelot mixing rule to calculate the Lennard-Jones (L–J) interactions between the water molecules and the carbon and hydrogen atoms of the nanoporous multilayered graphene. A timestep of 1 femtosecond was used to simulate the simulations for 6 nanoseconds. We used pppm style (particle-particle particle-mesh) solver to calculate the long-range electrostatic interactions [42]. The cutoff for the L–J interaction used in this work is 10 Å. The simulations are carried out using LAMMPS software [43] and we used VMD (visual molecular dynamics) software for visualization [44]. The water molecules are kept constrained using the SHAKE algorithm [45]. Nosé-Hoover thermostat [46], along with the canonical ensemble NVT is used in this study. To simulate the pressure-driven flow, the desired pressure is applied onto the piston (graphene sheet) at the feed region, and ambient pressure to the piston at the permeate region is applied [47].

In this study, a pressure of about 150 MPa is applied to the piston at the feed region. Although a pressure of 150 MPa is significantly higher than the typical desalination system, which is around a few MPa, previous studies have shown that the results will be valid at low pressures as well since the time scales for flow scale linearly with pressure applied [8]. As mentioned earlier, we have employed both functionalized and non-functionalized porous graphene in this study. It should be noted that the effective pore size decreases as the pore is functionalized.

## 3. Cumulative Molecule Passage and Occupancy

Figure 3 shows the cumulative water molecules passed through the non-functionalized and functionalized multilayered graphene membrane. SPC water model and TIP3P-EW water models showed quite similar higher water molecule passage, while OPC3 and TIP3P-FB showed lower water molecule passage in the non-functionalized multilayered membrane system. SPC water model showed around a 55% increase in the number of water molecules when compared to the TIP3P-FB water model. In the membrane system containing the pores functionalized with hydrogen atoms, the SPC water model showed around 120 water molecules being filtered, whereas the OPC3 water model filtered only around 7 water molecules. In the functionalized membrane system modeled using the OPC3 water model, the pore remained empty most of the time. This could be visualized easily in Figure 4.

In the case of the non-functionalized pore, the TIP3P-FB and SPC/E showed better water occupancy. The occupancy of water molecules inside the channel is determined by the local excess chemical potential [48]. Even though SPC and TIP3P-EW sowed relatively lower water molecules occupancy, these water models favored better transport of the water molecules as they show the relatively lesser fluctuation of water molecules inside the pore. A large part of the variation in the number of molecules that are filtered can be attributed to the partial charges of the atoms used in the models of water. As mentioned in a previous study [38], this can be viewed clearly when we compare the SPC and SPC/E, water models. These two water models differ only in their partial charges, and they show a variation of 40% in the number of water molecules filtered. Another key factor that affects the water transport through the pore is the energy constant (ε). This can be understood clearly by comparing the number of water molecules filtered in a functionalized membrane between the OPC3 water model and TIP3P-FB water models. The TIP3P-FB water model has a lower energy constant (ε) and a larger Van der Walls radius when compared to the OPC3 water model. In the case of the functionalized pore, TIP3P-FB showed a greater number of water molecules filtered when compared to that of the OPC3 water model, suggesting that the energy constant (ε) also plays a key role when the pore is very confined.

## 4. Free Energy of Occupancy Fluctuations

Figure 5 shows the occupancy fluctuation of the water molecules inside the non-functionalized and functionalized multilayered graphene pore. The detailed discussions regarding the free energy of occupancy fluctuations have been reported in many previous works [48,49,50]. In this work, we found that the maximum number of water molecules inside the channel is found in a non-functionalized membrane with a TIP3P-FB water model. In general, constant fluctuations of water molecules are seen in all cases. The constant fluctuation of the water molecules inside the channel shows that the average binding energy of water molecules inside the channel is unfavorable compared to bulk water. TIP3P-EW water model showed more fluctuations in the non-functionalized membrane. It also showed a very lower value of 15 for the most favorable number of atoms. The maximum value of 19 for the most favorable number of atoms is the TIP3P-FB water model. For the functionalized pore cases, most of the time, the pore remains empty. The maximum number of water molecules found in the functionalized pore is in the case of the SPC water model with 11 water molecules. The list of all water molecule occurrences and the instances inside the nanopore for both functionalized and non-functionalized cases using different water models is listed in Table 2.

## 5. Radial Distribution Function and Density

A particle’s probability of being found at a distance r away from a given reference particle is measured by RDF [51]. The radial distribution function inside the nanopore for the oxygen atoms to the first carbon atom of the pore is given in Figure 6. The RDF is computed for the same force cutoff distance used in the simulation. From Figure 6 of the non-functionalized pore, two atomic layers can be distinguished inside the channel from the two peaks in the radial distribution function plot [52]. The first distinctive peak occurs around 4 Å and the second distinctive peak occurs around 7.5 Å. For all water models, the position of peaks almost remains the same. The non-functionalized pore in Figure 6 also shows that the interaction between carbon atoms and oxygen atom of the TIP3P-EW water model is weak when compared to other water models. Hence, water molecules of the TIP3P-EW water model can slip freely through the multilayered graphene pore when compared to other water model cases.

The plot of the number density of the water molecules inside the nanopore is given in Figure 7. For the non-functionalized pore, we can see an increase in the density of the water molecules after the third layer of the porous graphene sheet. The water molecules enter the pore with large energy, this energy is subsequently lost as it crosses the third graphene layer, which results in the accumulation of a higher number of molecules in that region. Between the first and second layers of the multilayer, we can observe that molecules are mostly stagnated in the functionalized pore. It indicates that water molecules had difficulty overcoming the large energy barrier inside the pores.

## 6. Conclusions

In this study, we investigated the amount of variation in water molecules transported across functionalized and non-functionalized multilayered graphene using different water models. The five water models include the OPC3 water model, which claims to be more accurate in reproducing a comprehensive set of bulk properties when compared to water models of the same class such as TIP3P and SPC/E [5]. The other models are SPC, SPC/E, TIP3P-FB and TIP3P-EW. Different water models predict the different properties of water close to experimental results. So, in the transport of water through the nanopore, it is impossible to determine which water model best describes real water behavior [53]. However, based on the diffusion of water molecules in both functionalized and non-functionalized pores, TIP3P-FB predicts the diffusion of water molecules through multilayered porous graphene close to experimental results. For the non-functionalized pore water transport, we can say that the OPC3, TIP3P-FB and SPC/E water models predict close to real water diffusion. OPC3 and TIP3P-FB slightly predict the diffusion, while SPC/E slightly over-predicts the diffusion of water molecules. TIP3P-EW and SPC showed a similar higher number of transport of water molecules when compared with other water models used in this study. SPC and SPC/E water models that only differ in partial charges showed a 40% difference in the number of water molecules transported through the non-functionalized pore, which shows the significance of the water model’s partial charge. Additionally, by comparing OPC3 and TIP3P-FB in the functionalized pore, we can understand that the transport of water molecules in a very confined pore is affected by the value of the energy constant (ε) of the water model.

## Figures and Tables

**Figure 1 membranes-12-01109-f001:**
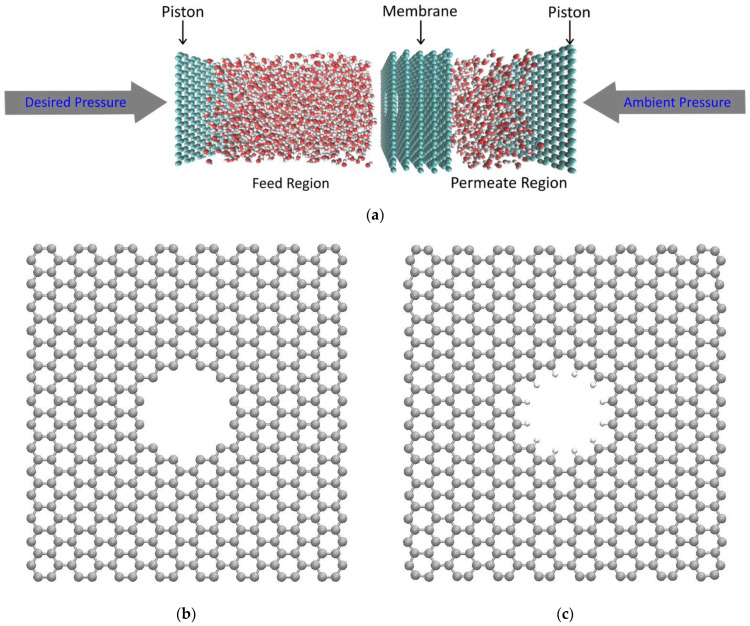
(**a**) Simulation domain, (**b**) porous graphene and (**c**) functionalized (hydrogenated) porous graphene.

**Figure 2 membranes-12-01109-f002:**
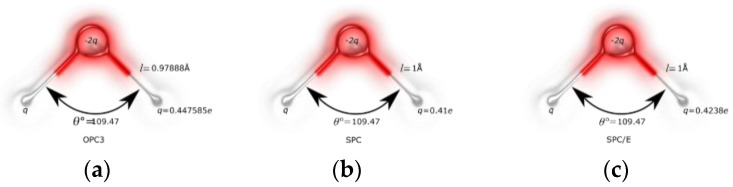
Different water models used in this study. (**a**) OPC3; (**b**) SPC; (**c**) SPC/E; (**d**) TIP3P−EW; (**e**) TIP3P−FB.

**Figure 3 membranes-12-01109-f003:**
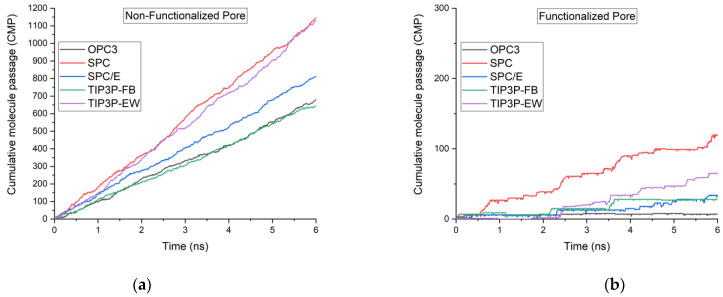
Cumulative water molecule passage through (**a**) multilayered non-functionalized porous graphene and (**b**) multilayered functionalized porous graphene using different water models.

**Figure 4 membranes-12-01109-f004:**
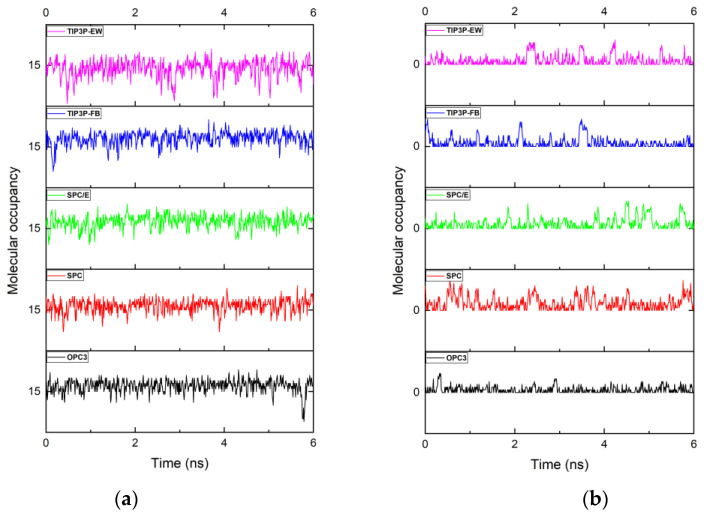
Number of water molecules inside the (**a**) multilayered non-functionalized porous graphene and (**b**) multilayered functionalized porous graphene using different water models at a given time.

**Figure 5 membranes-12-01109-f005:**
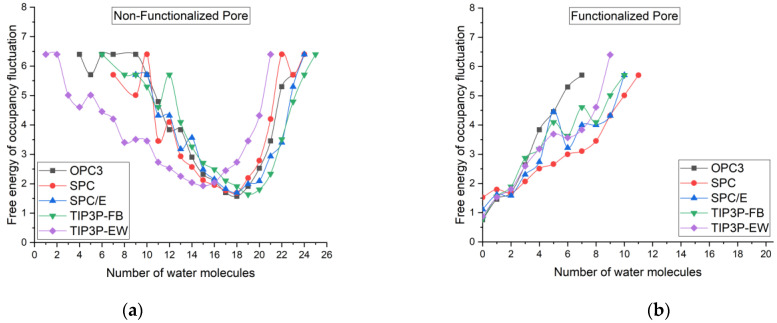
Free energy of occupancy fluctuations of water molecules inside (**a**) multilayered non-functionalized porous graphene and (**b**) multilayered functionalized porous graphene using different water models.

**Figure 6 membranes-12-01109-f006:**
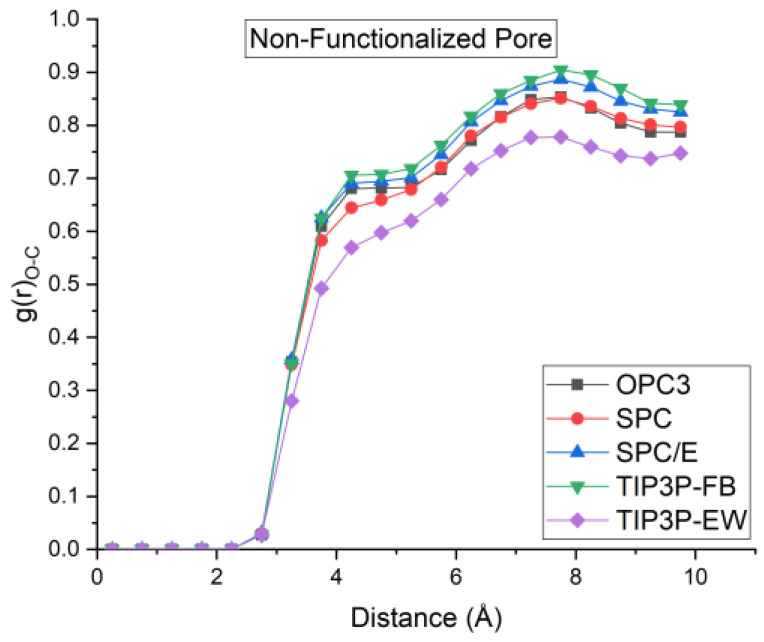
Radial distribution function of oxygen atoms of the water molecules with that of the carbon atoms inside the multilayered non-functionalized porous.

**Figure 7 membranes-12-01109-f007:**
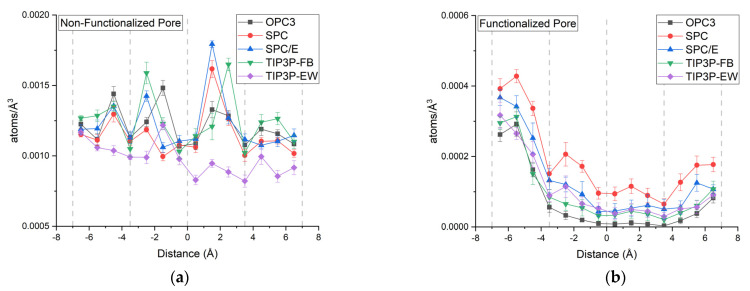
Number density of water molecules inside (**a**) non−functionalized and (**b**) functionalized pore (grey dashed lines represent the initial location of graphene layers).

**Table 1 membranes-12-01109-t001:** LJ parameters and atomic charges employed for water molecules in this work.

Water Model	σ_O_ (Å)	ε_O_ (kcal/mol)	q_H_ (e)	q_O_ (e)
SPC	3.16557	0.1554	0.41	−0.82
SPC/E	3.16557	0.1554	0.4238	−0.8476
TIP3P-FB	3.178	0.15587	0.41722	−0.84844
TIP3P-EW	3.188	0.102	0.415	−0.83
OPC3	3.17427	0.1634	0.447585	−0.89517

**Table 2 membranes-12-01109-t002:** List of water molecules and number of occurrences inside the pore for the 6 ns simulation.

Number of Water Molecules	Number of Occurrences									
	Non-Functionalized Pore					Functionalized Pore				
	OPC3	SPC	SPC/E	TIP3P-FB	TIP3P-EW	OPC3	SPC	SPC/E	TIP3P-FB	TIP3P-EW
0	0	0	0	0	0	282	131	197	275	250
1	0	0	0	0	1	140	100	119	128	130
2	0	0	0	0	1	111	113	123	91	99
3	0	0	0	0	4	43	76	60	34	45
4	1	0	0	0	6	13	49	39	25	25
5	2	0	0	0	4	7	42	7	10	15
6	1	0	0	1	7	3	30	24	16	17
7	1	2	0	0	9	2	27	11	6	13
8	0	0	0	2	20	0	19	11	10	6
9	1	4	2	2	18	0	8	8	4	1
10	2	1	2	3	19	0	4	2	2	0
11	5	19	8	6	39	0	2	0	0	0
12	13	10	8	2	48	0	0	0	0	0
13	13	32	25	10	63	0	0	0	0	0
14	33	46	17	23	78	0	0	0	0	0
15	59	72	50	40	87	0	0	0	0	0
16	74	85	70	50	78	0	0	0	0	0
17	110	102	97	73	52	0	0	0	0	0
18	125	111	110	89	39	0	0	0	0	0
19	89	67	82	117	19	0	0	0	0	0
20	48	37	74	99	8	0	0	0	0	0
21	19	9	32	58	1	0	0	0	0	0
22	3	1	20	18	0	0	0	0	0	0
23	2	2	3	5	0	0	0	0	0	0
24	0	1	1	2	0	0	0	0	0	0
25	0	0	0	1	0	0	0	0	0	0

## Data Availability

The data presented in this study are available on request from the corresponding author. The data are not publicly available due to privacy and ethical restrictions.

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
