# Peer review of "On the Choice of Different Water Model in Molecular Dynamics Simulations of Nanopore Transport Phenomena"

_membranes, 2022, doi:10.3390/membranes12111109_

Round 1
Reviewer 1 Report
Journal: Membranes
Manuscript ID: 1963585
The article entitled “On the choice of different water model in molecular dynamics simulations of nanopore transport phenomena” is based on molecular dynamics simulations dealing with the role of the water model on the results of water transport in graphene membrane. Carbon nanomaterials such as graphene nanomaterials, are of great importance since their adsorption performances are generally better than other conventional adsorbents. To observe the adsorption and to model this surface correctly, this work aims at studying the role of the water model on the transport properties of the graphene membrane. But this article relies only on models and calculations and never compares to experimental data that could help the reader to choose the model of water.
The text is clear, with logical progression but this work can be improved by taking into account the suggestions below.
Comments / suggestions
1. Recent literature on the role of water model in MD simulation should be cited (Dr Mejri in J. Mol. Liquid, 2022). The introduction can also be improved by citing some amazing experimental results in the field of nanopore for applications such as sensor, DNA sequencing etc (see works of Dr. Balme for instance).
2. In fig. 1, hydrogen atoms should be depicted in white (red is for oxygen)
3. Time for each simulations is quite low (6ns may be too short for describing appropriately the production runs) Authors should increase this time to see the evolution of the different data.
4. MD simulations are performed to obtain physical observables that should be averaged on time. Authors should thus give to reader table summarizing all the data with average and uncertainty in order to discuss the different results with more precision.
5. I did not see the size of the pore and the ions that are included in the bath??? Could the authors precise them. If no ion has been introduced in the bath, all the simulations should be performed again. The water molecule behaviors are highly sensitive to ions.
6. To understand the role of the water model, free energy profile of water molecule through the membrane should be estimated in order to better understand the filling of the pore.
7. The article remains very theoretical and far from the experimental reality. Could the author argue on experimental data to justify which water model should be the most appropriate for future?
article should be placed in major revision.
Sincelery yours
Reviewer 2 Report
The manuscript by C. Park, F. Robinson, and D. Kim is devoted to the investigation of water model on the results of water transport through nanopores in multilayered graphene. The proper choice of models in molecular dynamics simulation is a principal issue that determine the physical relevance and correctness of simulation results. Thus, such kind of methodological works are of great importance and interest.
The study presented seems well designed and accurately described and there are no comments on the data presented. However, the manuscript lacks global conclusion on what kind of water model is most appropriate for the simulation reported. Such conclusion should be the key point of such kind of investigation and its absence decreases value of the research.
I strongly advices authors to review their results and provide clear recommendation on what water model (or models) can be used for simulation of the systems presented. These recommendations should be supported by discussion why the chosen water model must be used.
Reviewer 3 Report
The water transport through nano porous multilayered graphene at 300k is investigated using molecular dynamics (MD) simulation with different water models in this study. The authors used functionalized and non-functionalized membranes along with 5 different 3-point rigid water models: SPC, SPC/E, TIP3P-FB, TIP3P-EW and OPC3 water models. However, I think the article has many problems and mistakes. The paper might not be published in the current journal. I suggest the following issues to be addressed.
1. Why do the authors need to conduct research on different types of water molecules passing through nano pores of functionalized and non-functionalized membranes? That would be better if they can add some practical significance of this research work.
2. In the Abstract, “This work also shows the significance of partial charge and energy constant (ε) of the water model used.” I don't think this article explains partial charge and energy constant (ε).
3. It is recommended to use more accurate potentials such as Buckingham potential for this work, because using the Lennard-Jones equation to calculate interatomic potentials usually causes relatively large errors.
4. Figure 3 shows the cumulative water molecules passed through the non-functionalized and functionalized multilayered graphene membrane. The curve in the figure looks chaotic, so it should be enlarged. More details for the CMP should be exhibited in the part of Figure 3.
5. In the Free energy of occupancy fluctuations, the authors stated “The maximum number of water molecules found in the functionalized pore is in the case of the SPC water model with 12 water molecules.” In Figure 5, we only found 11 water molecules in the functionalized pore in case of the SPC water, which is inconsistent with the description of the manuscript. Is it wrong?
6. On page 6, “the water molecules enter the pore with large energy, this energy is subsequently lost as it crosses the third graphene layer”. What is the distance range of the third layer of graphene in figure 7, the authors can accurately display each range of graphene in the figure.
7. It is recommended that different data maps be clearly marked in a figure.
8. Inconsistent citation format of references.
Round 2
Reviewer 1 Report
some language or typo mistakes persist in this revised version.
MD simulation runs are still too short in my opinion to draw a complete view of the role of water model on its behavior in confined situation.
Reviewer 2 Report
I suggest to publish the manuscript in the current state.
Author Response
We thank you for your constructive feedback. Your comments on our work have provided us with valuable insights to refine its contents and analysis.
Thank you.
Reviewer 3 Report
The authors improved the manuscript according to the my comments. I would recommend accept this manuscript after a minor revision. The topic is focused on the molecular models of water in MD simulations. Actually, there are many state-of-the-art desalination works based on MD simulations, e.g., Science Advances 6: eaba9471 (2020); Nature Materials 20: 1551-1558 (2021); et al. I suggest the authors add some comments on these publications in Introduction section.
